# Will Abusive Supervision Promote Subordinates’ Voluntary Learning Behavior?

**DOI:** 10.3390/bs12090317

**Published:** 2022-09-01

**Authors:** Zengrui Xiao, Ying Wang

**Affiliations:** 1School of International Education, Zhejiang Sci-Tech University, Hangzhou 310018, China; 2Institute of Innovation System Research, Institute of Scientific and Technical Information of Zhejiang Province, Hangzhou 310007, China

**Keywords:** abusive supervision, overt abusive supervision, covert abusive supervision, voluntary learning behavior, public self-consciousness

## Abstract

Abusive supervision was traditionally viewed as a unidimensional construct and found detrimental in various fields, while there may be subdimensions associating with different consequences. This study aims to justify two subdimensions of abusive supervision, namely overt abusive supervision and covert abusive supervision, and investigate their effects on subordinates’ voluntary learning behavior, with public self-consciousness as a moderator. Data was acquired from a sample of 443 employees from China through a two-wave survey, and hypotheses were tested by hierarchical regression analysis. The empirical results demonstrated that overt abusive supervision promotes subordinates’ voluntary learning behavior at lower levels of public self-consciousness and hinders it otherwise, while covert abusive supervision promotes subordinates’ voluntary learning behavior homogeneously at different levels of public self-consciousness. The results suggest that supervisors could be mean and critical when encouraging subordinates to improve themselves, with subordinates’ public self-consciousness taken into consideration. However, abusive supervision should never be overused, not only because it is unethical and detrimental in many other fields, but also because the abused subordinates may just be preparing for leaving.

## 1. Introduction

The topic of how supervisors interact with subordinates has attracted a great deal of academic attention, and substantial studies have been focused on abusive supervision in the past two decades [1,2]. Abusive supervision, defined as “subordinates’ perceptions of the extent to which supervisors engage in the sustained display of hostile verbal and nonverbal behaviors, excluding physical contact” [3] (p. 178), has been found to be detrimental in various fields, from subordinates’ psychological distress to work-related attitudes, behaviors and performances [4,5,6]. Only a handful of studies came to the counterintuitive conclusion that abusive supervision facilitates subordinates’ creativity, productivity, and proactive behavior [7,8,9]. The extant literature undervalues the bright side of abusive supervision, especially in the Asia-Pacific region, where power distance is higher and abusive supervision is more socially acceptable [10]. Exploring the possible positive consequences of abusive supervision will provide a more comprehensive understanding of the construct and enable supervisors to interact with subordinates more effectively. To address this issue, this study devotes to explore whether abusive supervision promotes subordinates’ voluntary learning behavior, which is “the activities in which individuals engage at their own volition, including those outside of company time, with an implicit assumption that the learning activity will yield dividends in the current role or in the future” [11] (p. 1105).

In the meantime, abusive supervision has been examined mostly as a unidimensional construct, however, it may be more complex than “abusive” or “not abusive” and involve several subdimensions associating with different consequences [12,13]. Even though Tepper had put forward a unidimensional definition in her seminal article, she also admitted that abusive supervision contains behaviors reflecting indifference as well as willful hostility [3]. Similarly, Zhang and Liu (2018) had proposed separating abusive supervision into covert behaviors and overt behaviors after a thorough review [10]. In addition, there are also some empirical evidences suggesting that the items in Tepper’s scale load onto two distinct factors in exploratory factor analysis [3,14]. As the forms, scopes, and magnitudes of abusive behaviors vary, they may exert different impacts on subordinates [10]. However, this notion has not gained enough attention in the previous studies, and there is still no generally accepted subdimensions of abusive supervision. To fulfil this gap, this study aims to identify two subdimensions of abusive supervision, namely overt abusive supervision and covert abusive supervision, and justify their differences by comparing their effects on subordinates’ voluntary learning behavior.

Furthermore, the effect of abusive supervision also varies in subordinates’ characteristics, such as negative reciprocity norm [14], conscientiousness [15], reasons for working [16], self-control capacity [17], emotional intelligence [18], cognitive ability [19], mindfulness [20], and primary psychopathy [21]. Apart from these characteristics, subordinates’ public self-consciousness, which is a trait “concerning the general awareness of the self as a social object” [22] (p. 523), may also play a role in the effects of abusive supervision. Abusive supervision embarrasses subordinates by damaging their self-image and social image, but different individuals may take it differently [23,24,25]. People high in public self-consciousness are more concerned about their social image [26,27], and are more sensitive to interpersonal mistreatment and more anxious about negative feedbacks from others [28,29]. Therefore, this study takes public self-consciousness as a moderator, to clarify the relationship between abusive supervision and voluntary learning behavior in more depth.

## 2. Theories and Hypotheses

### 2.1. The Theoretical Background of Abusive Supervision

Although there has long been doubts about the definition of abusive supervision, Tepper’s definition remains the most widely accepted one [3]. One major challenge is that abusive supervision is objective, while it has been conceptualized and operationalized as a subjective assessment of “subordinates’ perceptions” [1,12]. Literature replies to this query by claiming that the evaluation of supervisor’s behavior is contextualized and individualized, only when the abuse was recognized will it take effect on subordinates [1,14]. However, there are also individual differences in the evaluation of abusive suppression, which should be taken as moderating factors to avoid interference in the definition. Another challenge concerns the scope of abusive supervision [13]. To better understand its connotation and subdimensions, it is necessary to figure out what is not abusive supervision. First, abusive supervision focuses exclusively on hostility perpetrated by supervisors, which is different from workplace bullying [30]. Second, abusive supervision excludes other forms of hostility (e.g., physical or sexual), which is different from victimization [31]. Third, abusive supervision does not include behaviors without hostility, which is different from petty tyranny [32]. Therefore, this study modifies Tepper’s definition slightly to be “the extent to which supervisors engage in the sustained display of hostile verbal and nonverbal behaviors, excluding physical contact”, and pays cautious attention to the scope of abusive supervision when discussing its subdimensions.

To explain the effects of abusive supervision, various theoretical perspectives had been developed. For example, based on justice theory, it is argued that abusive supervision greatly damages subordinate’s perceived interactional justice, procedural justice, and distributive justice, which in turn decrease subordinate’s positive attitudes and behaviors [3,33]. From another perspective, based on psychological resistance theory, individuals are believed to strive to maintain personal control [34], and subordinates under abusive supervision are apt to engage in resistant behaviors to restore their personal autonomy [9]. Further, from a retaliation perspective and referencing the theory of displaced aggression, it is argued that subordinates will react to the aggression and hostility in abusive supervision by engaging in deviant behaviors directed at the supervisor, the organization, and other individuals [14,35].

However, all these perspectives presume abusive supervision to be counterproductive, and there are still no sound theories to explain the possible positive outcomes of abusive supervision. In this respect, Tepper et al., had proposed that psychological experiences such as attention, avoiding further hostility, desire to prove the supervisor wrong, and preparing for new employment may constitute a performance enhancing pathway [1]. Oh and Farh had put forward an integrative framework of emotional process theory to examine how and why individuals vary in their perceptions, experiences, and responses to abusive supervision over time [36]. Furthermore, Zhang and Liu (2018) had associated the word “abuse” to three words (pressure, aggression, and threat) [10], which can be linked to the three kinds of emotion (anger, fear, and sadness) noted by Oh and Farh (2017) [36]. First, abuse is a major source of pressure [13,37]. It not only brings about uncomfortable feelings, but also pushes subordinates to correct their mistakes if the pressure is well settled. Second, abuse is a typical kind of aggression, which usually calls for retaliation [14]. However, subordinates may also strive to avoid future aggressions by exhibiting positive behaviors. Third, abuse is a great threat, which normally damages subordinates’ work passion and well-being [36]. However, it may also drive subordinates to devote more to work efforts. Even though these insights are far from a solid theory, they have laid a shallow foundation for the mechanism of how abusive supervision can be positive.

### 2.2. The Subdimensions of Abusive Supervision

The legitimacy of taking abusive supervision as a unidimensional construct has been long doubted. It is realized, even when the construct was newly developed, that there are many distinct manifestations of abusive supervision, such as public criticism, rudeness, loud tantrums, inconsiderate actions, and coercion [38]. Later, Tepper had addressed the possibility that abusive supervision might be more complex than “abusive” or “not abusive”, and involve several subdimensions associating with different antecedents and consequences [13]. It also argued that subordinates’ attributions of supervisors’ abusive behaviors help to explain the differences in their reactions to abusive supervision. Following this logic, Liu et al., examined the contingent roles of subordinates’ attributions and found a stronger impact of abusive supervision on subordinates’ creativity when they attributed the abuse to supervisors’ injury initiation rather than performance promotion motives [39]. Furthermore, Burton et al., found three subdimensions of abusive supervision by linking it with subordinates’ attributions, namely internal, external, and relational [40]. However, it is noticeable that certain kinds of abusive behaviors are more likely to be attributed to injurious or internal motives than others. Rather than separating abusive supervision according to subordinates’ attributions, it is more reasonable to separate it according to supervisor’s motives.

In addition to the theoretical insights, the extant literature also provides some empirical evidences for this notion. Tepper had developed a 15-item scale of abusive supervision, which consists of 15 kinds of abusive behaviors [3]. By reanalyzing the data from previous studies, Mitchell and Ambrose had found that the fifteen indicators loaded onto two factors, which were labeled active-aggressive abusive behavior and passive–aggressive abusive behavior [3,14,41]. Regrettably, Mitchell and Ambrose claimed that only active-aggressive abusive behaviors capture the essence of abusive supervision and ignores the insightful differences between the two kinds of abusive behaviors [14]. Similar to the results of Mitchell and Ambrose, Zhang and Liu had proposed dividing abusive supervision into overt behaviors (e.g., yelling, public punishment) and covert behaviors (e.g., rude looks, ignoring someone, taking undue credit for what subordinates have done, withholding information from employees) [10]. In fact, the manifestations of active-aggressive abusive behaviors contain “ridicules”, “stupid”, “puts me down”, “negative comments” and “incompetent”, which are overt and humiliate subordinates publicly; while the manifestations of passive–aggressive abusive behaviors contain “invades my privacy”, “doesn’t give me credit”, “blames me”, “breaks promises” and “lies to me”, which are covert but may also infuriate subordinates greatly. Even though the passive–aggressive abusive behaviors are covert, they also focus on hostility perpetrated by supervisors, exclude physical hostility, and do not include behaviors without hostility. Therefore, we argue that they also capture the essence of abusive supervision. In line with this research, this study distinguished abusive supervision into two subdimensions: overt abusive supervision and covert abusive supervision. Overt abusive supervision refers to “the extent to which supervisors engage in the sustained display of overt hostile behaviors”, such as yelling, negative comments, and public humiliation, while covert abusive supervision refers to “the extent to which supervisors engage in the sustained display of covert hostile behaviors”, such as ignoring, lying, and taking undue credit.

### 2.3. The Effect of Abusive Supervision on Subordinates’ Voluntary Learning Behavior

As supervisors generally fail to self-criticize their misconducts and transform their manners, subordinates suffer from enduring abusive supervision and have to adapt themselves to satisfy supervisors or prepare for alternative choices [1,9]. To be more competitive, subordinates must keep updating their knowledge and abilities, and learning is one of the most effective ways that promotes such success. Traditionally, workplace learning relied upon formal training programs; however, constraints in time, budget, and geography hinder organizations to offer, and employees to attend, such programs [42]. Subordinates’ ongoing voluntary learning behavior, engaged at their own volition to improve themselves, may be more effective for their self-development and organizational performance [11]. Voluntary learning behavior is more self-directed and casual, and considered as a form of organizational citizenship behavior because it will improve organizational effectiveness if subordinates stay in the organization [43]. It may be facilitated by economic exchange or social exchange as other organizational citizenship behaviors, or simply out of an individual’s urgent need for job skills, love of knowledge, and self-actualization [11].

Even though there is a great deal of criticism on abusive supervision, it is also observed that such supervisors are also likely to provide subordinates feedback, more suitable assignments, and even career mentoring, which offer subordinates better opportunities to improve themselves [8,44]. Anyway, overt abusive supervision is straight-out and pushes subordinates to realize their mistakes and incompetence, which is somehow conducive to subordinates’ career growth [9,45]. Even when the negative comments are inappropriate, subordinates will then strive to prove their supervisor wrong by improving themselves [1,36]. Subordinates may also be aware that the task is of great importance and mistakes will not be tolerated, and try harder to complete the task to avoid future abuse [1,36,46]. Therefore, overt abusive supervision may urge subordinates to be more proactive to learn. Hypothesis H1 was proposed as below:

**Hypothesis** **H1.***Overt abusive supervision will be positively associated with subordinates’ voluntary learning behavior*.

Even though covert abusive supervision is less noticeable, subordinates will also recognize it over time. It is more insidious and likely to be attributed to hostile motives and damages subordinate-supervisor exchanges [25,39,47]. As their contribution is not being valued, subordinates are also apt to feel this is unfair [48,49,50]. What’s worse, subordinates suffer greatly from depression [3], anxiety [13], emotional exhaustion [51,52], and ego depletion [53]. Being unsatisfied and losing intrinsic enjoyment at work, subordinates will prepare for turnover [54]. To be competent for alternative job opportunities, they have to acquire more skills and engage in voluntary learning behavior. Therefore, covert abusive supervision may also drive subordinates to learn. Hypothesis H2 was proposed as below:

**Hypothesis** **H2.***Covert abusive supervision will be positively associated with subordinates’ voluntary learning behavior*.

### 2.4. The Moderating Effect of Public Self-Consciousness

Abusive supervision embarrasses subordinates by damaging their self-image and social image [24,25]. Individuals normally expect others to treat them fairly and avoid threatening behavior to their faces, but there is individual differences in this tendency and abusive supervision may not influence all subordinates in the same way [14,33]. Public self-consciousness characterizes the individual differences in the awareness of personal social image and the response to other peoples’ evaluations [22]. Those who are high in public self-consciousness, are concerned more about their social image, and are more likely to alter their actions to be socially desirable [23,28]. As they take their social image more seriously, they are also more sensitive to interpersonal rejection and more anxious about negative evaluations from others [26,55].

Overt abusive supervision embarrasses subordinates publicly and threatens their social image greatly, which normally generates various negative emotions [6]. Those who are low in public self-consciousness are less anxious about the negative feedbacks and will be less irritated [29,56]. Therefore, they will be able to deal with the experience more positively and engage in voluntary learning behavior to acquire more skills [9,39]. In contrast, those who are high in public self-consciousness are more likely to feel depressed and develop negative work-related attitudes [10,36,48]. To maintain their social image, they are more likely to blame their supervisor instead of admitting their own problems, and refuse to do as their supervisor requested [15,24]. Being infuriated, they will even confront their supervisor face to face, which also diminishes their efforts to improve themselves [5]. Therefore, those who are low in public self-consciousness will handle overt abusive supervision better and engage in more voluntary learning behavior. Hypothesis H3 was proposed as below:

**Hypothesis** **H3.***Public self-consciousness moderates the relationship between overt abusive supervision and subordinates’ voluntary learning behavior, such that the relationship is stronger at lower levels of public self-consciousness*.

Covert abusive supervision usually takes place unobtrusively, doing less harm to subordinates’ social image [24,39]. However, subordinates are not less irritated, because they can also feel the insult and threat. Those who are high in public self-consciousness are so sensitive to others’ evaluation that they may be bothered more by covert abusive supervision [27,28,56]. As covert abusive supervision is subtle and subordinates cannot outburst to relieve their anger, they will grow sad and experience emotional exhaustion [51,57], which damages their intrinsic enjoyment at work and self-efficacy to learn [25,49,58]. In contrast, those who are low in public self-consciousness may be less annoyed. They feel less sorry for themselves and will not confront with supervisors for the time being [26,36]. Instead, they will focus on the urgent need for self-improvement and engage more in voluntary learning behavior [1,45]. Therefore, the effect of covert abusive supervision will be stronger for those who are low in public self-consciousness. Hypothesis H4 was proposed as below:

**Hypothesis** **H4.***Public self-consciousness moderates the relationship between covert abusive supervision and subordinates’ voluntary learning behavior, such that the positive relationship is stronger at lower levels of public self-consciousness*.

The research framework and hypotheses are presented in Figure 1.

## 3. Method

### 3.1. Sample and Procedures

A two-wave survey was carried out in 27 companies from China. These companies engage in various industries, such as textile, energy, real estate, finance, and information technology. Overt abusive supervision and covert abusive supervision were measured in the first stage, public self-consciousness and voluntary learning behavior were measured ten to fifteen days later, while demographic variables were recorded in both stages. In the front of the questionnaire, a brief introduction was given to inform the participants that the survey was anonymous and the results will only be used for academic purposes. To ensure anonymity, we did not ask for email address or other personal information to match the data. At the first stage, an online questionnaire was sent to the companies’ WeChat groups, and anyone accomplished the survey got a random retribution of ¥ 0.01 to ¥ 10.00. At the second stage, the respondents were asked to check and fill in the amount of their retributions, which will be used to match the data. Demographic variables will be used for double check. A total of 571 respondents completed the questionnaire in the first stage, and 443 of them completed the questionnaire in the second stage (effective rate of return = 77.6%). Among the respondents, 202 were female (45.6%) and 241 were male (54.4%). As for age, 100 were 25 or younger (22.6%), 147 were 26–35 (33.2%), 121 were 36–45 (27.3%), and 75 were 46 or older (16.9%). As for education level, 84 had a college degree or lower (19.0%), 166 had a bachelor degree (37.5%), 137 had a master degree (30.9%), and 56 had a doctoral degree (12.6%). As for work tenure, 110 had been working in the firm for less than 1 year (24.8%), 136 had been working in the firm for 1–3 years (30.7%), 116 had been working in the firm for 4–5 years (26.2%), and 81 had been working in the firm for 6 years or more (18.3%).

### 3.2. Measures

To test the hypotheses, existing scales developed and validated by previous studies were used to measure the four variables. As the survey was carried out in China, the items were translated from English into Chinese through the back-translation process until no discrepancy between the original items and the translations exist [59].

Abusive supervision: Mitchell and Ambrose identified two factors in Tepper’s 15-item scale, with five items loaded on the first factor, five items loaded on the second factor, and the other five items cross loaded on both factors [3,14]. The five items of the active-aggressive factor were used to measure overt abusive supervision [14]. One sample item is “My supervisor ridicules me.” The Cronbach’s α was 0.82 in this study. The five items of the passive–aggressive factor were used to measure covert abusive supervision [14]. One sample item is “My supervisor doesn’t give me credit for jobs requiring a lot of effort”. The Cronbach’s α was 0.85 in this study. A five-point Likert scale with anchors of frequency ranging from 1 (never) to 5 (very often) was used for these two variables.

Public self-consciousness: Public self-consciousness was measured with Fenigstein et al.’s seven-item scale, which was widely accepted [22]. One sample item is “I’m concerned about my style of doing things”. The Cronbach’s α was 0.87 in this study. A five-point Likert scale with anchors of agreement ranging from 1 (strongly disagree) to 5 (strongly agree) was used.

Voluntary learning behavior: Voluntary learning behavior was measured with Walumbwa et al.’s seven-item scale [11]. The referent was modified from “This person” to “I”. One sample item is “I ask for feedback on my performance”. The Cronbach’s α was 0.88 in this study. A five-point Likert scale with anchors of frequency ranging from 1 (never) to 5 (very often) was used.

Control variables: As suggested by previous studies, gender, age, education, and tenure were chosen as control variables [3,11,22].

## 4. Results

### 4.1. Reliabilities and Validities

Firstly, as reported previously, the Cronbach’s alpha of the variables ranged from 0.82 to 0.88, which indicates the internal reliabilities for all scales were acceptable. Secondly, confirmatory factor analysis of the four-factor model showed a good fit with the data (χ^2^ = 399.893, DF = 246, χ^2^/DF = 1.626, *p* < 0.001, SRMR = 0.041, GFI = 0.928, CFI = 0.964, RMSEA = 0.038), and the results were presented in Table 1. All the standardized regression weights (SRW) were strong and significant, supporting the items as indicators for the underlying constructs. Thus, the convergent validity was confirmed. Thirdly, the composite reliabilities (CR) of all variables were larger than 0.8, the average variances extracted (AVE) were larger than or very close to 0.5, and the square roots of AVE of all variables were larger than the corresponding inter-construct correlations. Thus, the composite reliability and discriminant validity were also confirmed. Fourthly, a longitudinal design was applied to minimize the common method bias in the data, and Harman’s single-factor test was conducted to check how severe the problem was. The result of exploratory factor analysis showed that the items loaded onto four factors, explained 59.0% of the variances in total, and the largest factor explained only 16.8% of the variances, which is lower than the threshold value of 40% and indicates that the common method bias was not severe.

### 4.2. Descriptive Analyses

The results of descriptive statistics, correlation matrix, and the square roots of AVE were presented in Table 2. For the binary variable of gender, female was coded as 0 and male was coded as 1. Overt abusive supervision was positively correlated with covert abusive supervision (r = 0.412, *p* < 0.01), but the coefficient is much less than 0.7. As can be seen in the validity analyses, the discriminant validity is also enough. Therefore, it is appropriate to distinguish abusive supervision into the two subdimensions. Meanwhile, voluntary learning behavior is positively correlated with both overt abusive supervision (r = 0.105, *p* < 0.05) and covert abusive supervision (r = 0.251, *p* < 0.01), and negatively correlated with public self-consciousness (r = −0.106, *p* < 0.05). These results provide preliminary support for the hypotheses.

### 4.3. Hypothesis Testing

The hypotheses were then tested with hierarchical regression analysis, with voluntary learning behavior as the dependent variable. Firstly, only the control variables were added to build Model 1. Secondly, on the basis of Model 1, the independent variables, overt abusive supervision and covert abusive supervision, and the moderator, public self-consciousness, were added to build Model 2. Thirdly, on the basis of Model 2, the interaction terms, OAS × PSC and CAS × PSC, were added to build Model 3. The interaction terms were computed by multiplying the two variables’ arithmetic means in advance. The variance inflation factors (VIF) for all models were checked and all the VIF are within the acceptable range (0 < VIF < 2), indicating that the multicollinearity was not severe. The regression coefficients, F-score and change in R^2^ were analyzed to judge whether the effects were significant. The results of the regression analysis were presented in Table 3.

As presented in Table 3, the results of Model 1 showed that gender (β = 0.122, *p* < 0.05) and education (β = 0.091, *p* < 0.05) were significantly related to subordinates’ voluntary learning behavior, while age (β = −0.001, *p* > 0.05) and tenure (β = 0.022, *p* > 0.05) were not. Therefore, male subordinates and those who have higher education are more likely to engage in voluntary learning behavior.

The results of Model 2 showed that the regression coefficient for overt abusive supervision (β = 0.010, *p* > 0.05) was not significant while the regression coefficient for covert abusive supervision (β = 0.243, *p* < 0.001) was positive and significant. Therefore, Hypothesis H1 was not supported and Hypothesis H2 was supported.

The regression results of Model 3 showed that the regression coefficient of OAS × PSC (β = −0.182, *p* < 0.001) was negative and significant, while the regression coefficient of CAS × PSC (β = −0.047, *p* > 0.05) was not significant.

To illuminate the moderating effects, the samples were separated into two groups, the low public self-consciousness group (mean—S.D.) and the high public self-consciousness group (mean + S.D.). The regression results were presented in Figure 2 and Figure 3. As presented in Figure 2, overt abusive supervision was negatively associated with subordinates’ voluntary learning behavior at higher levels of public self-consciousness and positively associated with subordinates’ voluntary learning behavior at lower levels of public self-consciousness. Thus, Hypothesis H3 was supported. As presented in Figure 3, covert abusive supervision was positively associated with subordinates’ voluntary learning behavior for both groups and the difference was not significant. Thus, Hypothesis H4 was not supported.

## 5. Discussion

### 5.1. Theoretical Insights

This study aimed to identify the subdimensions of abusive supervision and investigate their effects on subordinates’ voluntary learning behavior, comprising the moderating effects of public self-consciousness. Four hypotheses had been put forward, and two of them were supported. Some novel and inspiring insights can be drawn from these results.

Unexpectedly, Hypothesis H1 was not supported. Overt abusive supervision is not always positively associated with subordinates’ voluntary learning behavior. As illuminated in the moderating effect of public self-consciousness, overt abusive supervision promotes subordinates’ voluntary learning behavior at lower levels of public self-consciousness, which is consistent with our hypothesis. However, it is unexpected that overt abusive supervision hinders subordinates’ voluntary learning behavior at higher levels of public self-consciousness. As the two groups are mixed up, the relationship between overt abusive supervision and subordinates’ voluntary learning behavior appears to be non-significant.

As expected, Hypothesis H2 was supported. Covert abusive supervision is positively associated with subordinates’ voluntary learning behavior. The deleterious consequences of abusive supervision have been widely recognized in the literature [5,10], while this study provides new evidence for its positive outcomes. Of course, it should be noted that these subordinates are probably engaging in voluntary learning behavior out of a desire for alternative job opportunities. However, it may also contribute to the organizational performance to some extent, since some of them will stay when they are self-improved and have earned respect in the current organization [11,42].

It is also worth to highlighting that the positive effects of abusive supervision are quite limited. Even in the domain of subordinates’ voluntary learning behavior, negative and positive effects coexist. The negative pathway carries a much stronger effect than the positive pathway in many other cases [1,5]. Meanwhile, the effects of overt abusive supervision and covert abusive supervision on subordinates’ voluntary learning behavior are different, which provides further evidence for the distinction between the two subdimensions.

Hypothesis H3 was also supported. However, slightly differently from what was hypothesized, overt abusive supervision hinders subordinates’ voluntary learning behavior at higher levels of public self-consciousness, instead of positively and is weaker than that at lower levels of public self-consciousness. The possible reason may be that those who are high in public self-consciousness are so sensitive to the public humiliation of overt abusive supervision that they tend to feel deeply irritated and depressed [10,36]. Spending too much effort on fighting supervisors or recovering from self-pity damages their passion for voluntary learning behavior [5,15]. What’s worse, as they tend to confront with supervisors, supervisors may refuse to provide them developmental feedbacks, suitable assignments, and career mentoring [8,44].

Hypothesis H4 was not supported. Unexpectedly, covert abusive supervision promotes subordinates’ voluntary learning behavior homogeneously at different levels of public self-consciousness. The possible reason may be that even though covert abusive supervision is offensive, it takes place privately and does less harm to subordinates’ social image. Therefore, even those who are high in public self-consciousness will not be embarrassed too much. However, being infuriated by the insult of covert abusive supervision, all subordinates will be eager to improve themselves so that they will be better prepared to turnover [51,54]. Therefore, no matter they are high or low in public self-consciousness, subordinates will engage in more voluntary learning behavior in response to covert abusive supervision.

### 5.2. Practical Implications

This study is not trying to defend abusive supervision, nor is it intended to advocate supervisors to be abusive. Abusive supervision is unethical after all, let alone that the positive aspects of abusive supervision are rather insignificant compared with other deleterious consequences. However, just as Ferris et al., suggested, supervisors need to engage in coercive power when subordinates were unable or unmotivated to take responsibility [60]. The recommendation that can be made is when supervisors are trying to encourage subordinates to improve themselves, through voluntary learning behavior for example, they can try to be mean and critical. To achieve a positive outcome, supervisors must pay attention to subordinates’ public self-consciousness and avoid humiliating those who are of high public self-consciousness publicly. Meanwhile, it is always wise to remember that, subordinates may engage in voluntary leaning behavior just to improve themselves for alternative job opportunities. If supervisors fail to control their manners, the least thing they need to do would be convincing subordinates that the abuse is not out of hostility but for their own good.

### 5.3. Limitations and Future Research Directions

This study is not without limitations. First, this study had modified Tepper’s definition of abusive supervision from “subordinates’ perception” to an objective state [3], but still measured it with subordinates’ self-reports as most previous studies. As abusive supervision may not always align with subordinates’ perceptions, measuring it with organizational records or evaluations from a familiar third party may be more accurate. Second, this study has focused on the effect of abusive supervision on subordinates’ voluntary learning behavior, and neglects the effects on subordinates’ psychological distress, interaction with other employees, organizational culture, and corporation image. The limited productive effect on employee’s self-improvement is probably not able to cover the counterproductive effects in other aspects. It will be more inspiring to investigate the consequences of abusive supervision in an integrated framework and illuminate under what conditions will the productive effects exceed the counterproductive effects. Third, the survey was conducted only in China, where power distance is much higher than the West. Individuals from China may more easily accept abusive supervision and respond more positively. Therefore, culture may have a great impact on subordinates’ perception of and response to abusive supervision. Cross-cultural comparison studies should be done in the future to examine whether the effects of overt and covert abusive supervision can be generalized in other cultures. Fourth, this study has demonstrated the positive effects of abusive supervision on subordinates’ voluntary learning behavior, further investigations considering the positive outcomes for individuals, teams, and organizations in other fields under certain conditions will be interesting and inspiring.

## 6. Conclusions

In summary, this study has demonstrated that overt abusive supervision promotes subordinates’ voluntary learning behavior at lower levels of public self-consciousness and hinders it otherwise, while covert abusive supervision promotes subordinates’ voluntary learning behavior homogeneously at different levels of public self-consciousness. This study contributes to the literature by identifying the two subdimensions of abusive supervision, discovering the positive effect of abusive supervision on subordinates’ voluntary learning behavior, and clarifying the moderating effect of public self-consciousness.

## Figures and Tables

**Figure 1 behavsci-12-00317-f001:**
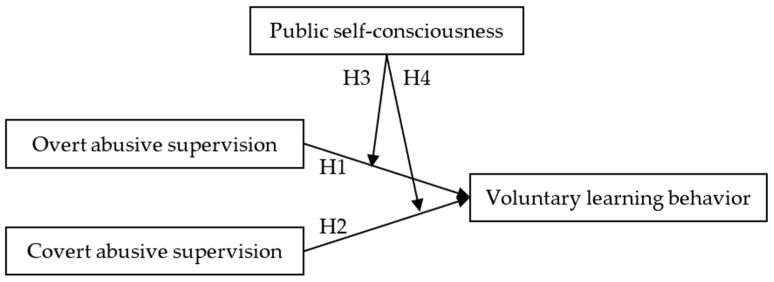
Research framework and hypotheses.

**Figure 2 behavsci-12-00317-f002:**
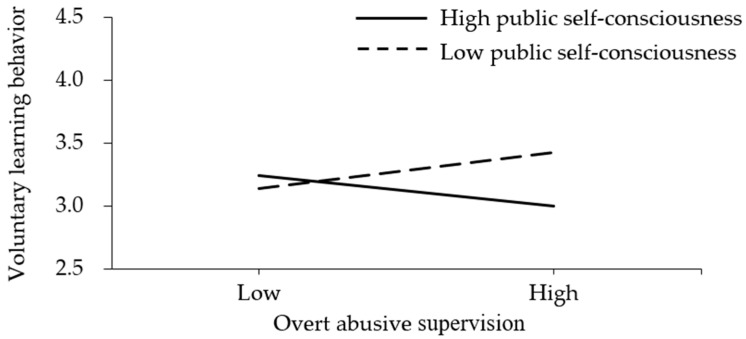
The moderating effect of public self-consciousness on the relationship between overt abusive supervision and subordinates’ voluntary learning behavior.

**Figure 3 behavsci-12-00317-f003:**
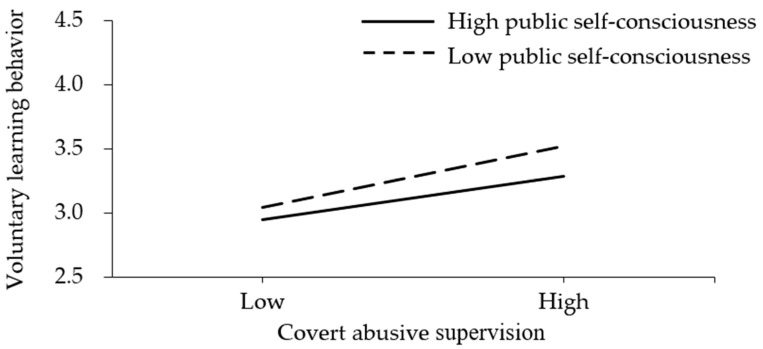
The moderating effect of public self-consciousness on the relationship between covert abusive supervision and subordinates’ voluntary learning behavior.

**Table 1 behavsci-12-00317-t001:** Measurements and validities.

Variables	Items	SRW	CR	AVE
OAS	A1. My supervisor ridicules me.	0.68	0.814	0.468
	A2. My supervisor tells me my thoughts or feelings are stupid.	0.66		
	A3. My supervisor puts me down in front of others.	0.66		
	A4. My supervisor makes negative comments about me to others.	0.74		
	A5. My supervisor tells me I’m incompetent.	0.67		
	Source: Tepper [3], Mitchell and Ambrose [14]			
CAS	P1. My supervisor invades my privacy.	0.76	0.853	0.536
	P2. My supervisor doesn’t give me credit for jobs requiring a lot of effort.	0.68		
	P3. My supervisor blames me to save himself/herself embarrassment.	0.75		
	P4. My supervisor breaks promise he/she makes.	0.75		
	P5. My supervisor lies to me.	0.72		
	Source: Tepper [3], Mitchell and Ambrose [14]			
PSC	S1. I’m concerned about my style of doing things.	0.67	0.873	0.496
	S2. I’m concerned about the way I present myself.	0.75		
	S3. I’m self-conscious about the way I look.	0.74		
	S4. I usually worry about making a good impression.	0.73		
	S5. One of the last things I do before I leave my house is look in the mirror.	0.67		
	S6. I’m concerned about what other people think of me.	0.72		
	S7. I’m usually aware of my appearance.	0.65		
	Source: Fenigstein et al. [22]			
VLB	L1. I ask for feedback on my performance.	0.70	0.875	0.501
	L2. I rely on outdated information or ideas. *	0.65		
	L3. I actively review my own progress and performance.	0.80		
	L4. I do my work without stopping to consider all the information team members have. *	0.75		
	L5. I regularly take time to figure out ways to improve my work performance.	0.67		
	L6. I ignore feedback from others in the company. *	0.69		
	L7. I ask for help from others in the company when something comes up that I don’t know how to handle.	0.68		
	Source: Walumbwa et al. [11]			

* Reverse scored. OAS represents overt abusive supervision, CAS represents covert abusive supervision, PSC represents public self-consciousness, VLB represents voluntary learning behavior.

**Table 2 behavsci-12-00317-t002:** Means, standard deviations, and correlations.

Variables	Mean	S.D.	1	2	3	4	5	6	7	8
1. Gender	0.54	0.499								
2. Age	2.39	1.015	0.085							
3. Education	2.37	0.932	−0.033	0.056						
4. Tenure	2.38	1.049	0.068	0.041	0.022					
5. OAS	2.83	0.856	0.011	0.003	−0.020	0.115 *	(0.684)			
6. CAS	3.01	0.899	0.076	−0.062	−0.054	0.051	0.412 **	(0.733)		
7. PSC	3.16	0.889	0.003	−0.152 **	0.029	−0.109 *	0.039	−0.023	(0.704)	
8. VLB	3.20	0.856	0.120 *	0.015	0.087	0.032	0.105 *	0.251 **	−0.106 *	(0.708)

* *p* < 0.05, ** *p* < 0.01 (two-tailed). Diagonal elements are the square root of AVE, whereas off-diagonal values are inter-construct correlations.

**Table 3 behavsci-12-00317-t003:** Results of regression analysis for voluntary learning behavior.

Variables	Model 1		Model 2		Model 3	
	b	β	b	β	b	β
Constant	2.850		2.469		2.446	
Control variables						
Gender	0.209	0.122 *	0.182	0.106 *	0.156	0.091 *
Age	−0.001	−0.001	−0.001	−0.001	0.001	0.001
Education	0.083	0.091 *	0.098	0.107 *	0.095	0.103 *
Tenure	0.018	0.022	−0.001	−0.002	0.001	0.001
Independent variables						
OAS			0.010	0.010	0.014	0.014
CAS			0.231	0.243 ***	0.239	0.251 ***
Moderator						
PSC			−0.100	−0.104 *	−0.097	−0.101 *
Interactions						
OAS × PSC					−0.154	−0.182 ***
CAS× PSC					−0.040	−0.047
F-score _(df1, df2)_	2.609_(4, 438)_ *		6.531_(7, 435)_ ***		7.567_(9, 433)_ ***	
R^2^	0.023		0.095		0.136	
Adjusted R^2^	0.014		0.081		0.118	
∆F-score _(df1, df2__)_			11.510_(3, 435__)_ ***		10.224_(2, 433__)_ ***	
∆R^2^			0.072		0.041	

* *p* < 0.05, *** *p* < 0.001; b is unstandardized beta, β is standardized beta.

## Data Availability

Data are available on request from the authors.

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
