# Peer review of "Will Abusive Supervision Promote Subordinates’ Voluntary Learning Behavior?"

_behavsci, 2022, doi:10.3390/bs12090317_

Round 1

Reviewer 1 Report

Title: Will Abusive Supervision Promote Subordinates’ Voluntary 2 Learning Behavior?

1.      Aim, Main contributions, and strengths of the study.

This study aims to justify two subdimensions of abusive supervision, namely overt abusive supervision, and covert abusive supervision, and investigate their effects on subordinates’ voluntary learning behavior, with public self-consciousness as a moderator.

The main contribution of the study are that:

·        overt abusive supervision promotes subordinates’ voluntary learning behavior at lower levels of public self-consciousness and hinders it otherwise, while covert abusive supervision promotes subordinates’ voluntary learning behavior homogeneously at different levels of public self-consciousness.

·         The results also suggest that supervisors could be mean and critical when encouraging subordinates to improve themselves, with subordinates’ public self-consciousness taken into consideration.

·        Abusive supervision should never be overused, not only because it is unethical and detrimental in many other fields, but also because it may cause the abused subordinates to leave the establishment.

2.      General concept comments
Areas of weakness: To my mind the study seems to suggest that abusive supervision may lead to employee’s self-improvement, although this seem to be the case, however the negative aspects of organizational culture and image even after a self-improvement is not taken into consideration e.g., the likelihood of the employee him/her self-adopting the same attitude towards other subordinates. Also, the impact of this on the corporations image particularly in this era of social media and smart phones. Also the study is focused in China, however one wonders if this study is conducted elsewhere if similar results would be obtained if not why?

3.      The testability of the hypothesis, methodological inaccuracies, missing controls, etc.

Through a two-wave survey, data from a sample of Chinese employees was collected, and hierarchical regression analysis was used to assess the hypotheses. The test used is accurate because, after controlling for all other variables, hierarchical regression analysis may be used to determine whether certain variables of interest account for a statistically significant portion of the variance in your dependent variables (DVs). The researchers were able to run regressions on several various DVs and compare the outcomes for each DV thanks to the utilization of this approach in this investigation.

4.      Review: The topic was adequately covered, the topic is contemporary and relevant, the authors identified a gap in literature which is firstly distinguishing between overt and covert abusive supervision with public self-consciousness as a mediating factor. Secondly the fact that overt abusive supervision promotes subordinates’ voluntary learning behavior at lower levels of public self-consciousness and hinders it otherwise, while covert abusive supervision promotes subordinates’ voluntary learning behavior homogeneously at different levels of public self-consciousness.

5.      Referencing: This was not understood by me, I am used to references being in alphabetical order however this was not the case here. Apart from that the article was well referenced.

6.      Specific comments: No specific comments different from above. Overall the study is recommended for publication, authors are recommended for a good work and the editorials should be further checked.

The article is well presented, articulated and relevant. It is scholarly and contributes significantly  to the field of study.

The reference section is not arranged alphabetically. I am not sure if this is the  requirement of the journal.

I recommend the article for publication after  editing for grammar and comprehension. All areas to be edited is highlighted in the article with comments.

Author Response

Thank you so much for your constructive comments! We believe that we have been able to address your comments and suggestions, and that our paper has been substantially improved thanks to you. Below, we will indicate how we have responded to each of your comments.

Comment 1:

  1. Aim, main contributions, and strengths of the study.

This study aims to justify two subdimensions of abusive supervision, namely overt abusive supervision, and covert abusive supervision, and investigate their effects on subordinates’ voluntary learning behavior, with public self-consciousness as a moderator.

The main contribution of the study are that:

  • Overt abusive supervision promotes subordinates’ voluntary learning behavior at lower levels of public self-consciousness and hinders it otherwise, while covert abusive supervision promotes subordinates’ voluntary learning behavior homogeneously at different levels of public self-consciousness.
  • The results also suggest that supervisors could be mean and critical when encouraging subordinates to improve themselves, with subordinates’ public self-consciousness taken into consideration.
  • Abusive supervision should never be overused, not only because it is unethical and detrimental in many other fields, but also because it may cause the abused subordinates to leave the establishment.

Response:

Thank you so much for your affirmation, it is of great help to our confidence in our theoretical contribution.

Comment 2:

  1. General concept comments

Areas of weakness: To my mind the study seems to suggest that abusive supervision may lead to employee’s self-improvement, although this seem to be the case, however the negative aspects of organizational culture and image even after a self-improvement is not taken into consideration e.g., the likelihood of the employee him/her self-adopting the same attitude towards other subordinates. Also, the impact of this on the corporations image particularly in this era of social media and smart phones. Also the study is focused in China, however one wonders if this study is conducted elsewhere if similar results would be obtained if not why?

Response:

Thank you so much for this constructive comment. First, we totally agree that the negative aspects of organizational culture, the attitude and behavior towards other subordinates, and the impact on the corporation image are also important aspects of abusive supervision and should be taken into consideration. For the concision of theory, we have focused on the effect of abusive supervision on subordinates’ voluntary learning behavior, but we also realize that this is not the whole picture, that’s why we have argued that “However, abusive supervision should never be overused, not only because it is unethical and detrimental in many other fields, but also because the abused subordinates may just be preparing for leaving.” We will discuss more about this problem in the limitation. Second, it is true that the results may not be the same if the study was conducted in other culture, that’s why we have emphasized the context of Chinese culture in the introduction and propose to conduct further studies in other culture in the limitation. We will discuss more about this problem in the limitation.

Comment 3:

  1. The testability of the hypothesis, methodological inaccuracies, missing controls, etc.

Through a two-wave survey, data from a sample of Chinese employees was collected, and hierarchical regression analysis was used to assess the hypotheses. The test used is accurate because, after controlling for all other variables, hierarchical regression analysis may be used to determine whether certain variables of interest account for a statistically significant portion of the variance in your dependent variables (DVs). The researchers were able to run regressions on several various DVs and compare the outcomes for each DV thanks to the utilization of this approach in this investigation.

Response:

Thank you so much for your affirmation.

Comment 4:

  1. Review: The topic was adequately covered, the topic is contemporary and relevant, the authors identified a gap in literature which is firstly distinguishing between overt and covert abusive supervision with public self-consciousness as a mediating factor. Secondly the fact that overt abusive supervision promotes subordinates’ voluntary learning behavior at lower levels of public self-consciousness and hinders it otherwise, while covert abusive supervision promotes subordinates’ voluntary learning behavior homogeneously at different levels of public self-consciousness.

Response:

Thank you so much for your affirmation.

Comment 5:

  1. Referencing: This was not understood by me, I am used to references being in alphabetical order however this was not the case here. Apart from that the article was well referenced.

Response:

Thank you so much for this comment. We have prepared the manuscript with the template provided by the journal, which requires the references to be numbered.

Comment 6:

  1. Specific comments: No specific comments different from above. Overall, the study is recommended for publication, authors are recommended for a good work and the editorials should be further checked.

The article is well presented, articulated and relevant. It is scholarly and contributes significantly to the field of study.

The reference section is not arranged alphabetically. I am not sure if this is the requirement of the journal.

I recommend the article for publication after editing for grammar and comprehension. All areas to be edited is highlighted in the article with comments.

Response:

Thank you so much for your meticulous instruction. We have checked the grammar and expression throughout the manuscript. Thank you for your help!

Reviewer 2 Report

The authors have been able to develop a work that facilitates its reading and that allows them to communicate their academic and practical contribution.

Author Response

Comments and Suggestions for Authors:

The authors have been able to develop a work that facilitates its reading and that allows them to communicate their academic and practical contribution.

Response:

Thank you so much for your positive evaluation. You will never know how much your encouragement meant to us. Even though you didn’t propose any further revisions, it will be a great honor and help to us that if you would review the revised manuscript and comment about whether the revisions are appropriate. Thank you again for your time and effort.

Reviewer 3 Report

A very interesting and well-presented article. 

Author Response

Comments and Suggestions for Authors:

A very interesting and well-presented article.

Response:

Thank you so much for your positive evaluation. You will never know how much your encouragement meant to us. Even though you didn’t propose any further revisions, it will be a great honor and help to us that if you would review the revised manuscript and comment about whether the revisions are appropriate. Thank you again for your time and effort.